# Tumor-Infiltrating Lymphocyte Level Consistently Correlates with Lower Stiffness Measured by Shear-Wave Elastography: Subtype-Specific Analysis of Its Implication in Breast Cancer

**DOI:** 10.3390/cancers16071254

**Published:** 2024-03-22

**Authors:** Na Lae Eun, Soong June Bae, Ji Hyun Youk, Eun Ju Son, Sung Gwe Ahn, Joon Jeong, Jee Hung Kim, Yangkyu Lee, Yoon Jin Cha

**Affiliations:** 1Department of Radiology, Gangnam Severance Hospital, Yonsei University College of Medicine, Seoul 06273, Republic of Korea; enrlove@yuhs.ac (N.L.E.); jhyouk@yuhs.ac (J.H.Y.); ejsonrd@yuhs.ac (E.J.S.); 2Institute of Breast Cancer Precision Medicine, Yonsei University College of Medicine, Seoul 06273, Republic of Korea; mission815815@yuhs.ac (S.J.B.); asg2004@yuhs.ac (S.G.A.); gsjjoon@yuhs.ac (J.J.); ok8504@yuhs.ac (J.H.K.);; 3Department of Surgery, Gangnam Severance Hospital, Yonsei University College of Medicine, Seoul 06273, Republic of Korea; 4Division of Medical Oncology, Department of Internal Medicine, Yonsei University College of Medicine, Seoul 06273, Republic of Korea; 5Department of Pathology, Gangnam Severance Hospital, Yonsei University College of Medicine, Seoul 06273, Republic of Korea

**Keywords:** breast neoplasms, elastography, tumor infiltrating lymphocyte, pathology

## Abstract

**Simple Summary:**

Tumor stiffness in breast cancer displays varied clinical implications depending on the tumor subtype, with higher stiffness indicating a more aggressive tumor biology particularly in hormone receptor-positive and HER2-negative breast cancer. This study investigated the relationship between tumor stiffness, measured by shear-wave elastography, and clinicopathologic parameters such as the tumor-infiltrating lymphocytes (TIL) levels in 803 breast cancer patients across different subtypes. The results showed that higher tumor stiffness is associated with more aggressive tumor features, especially in hormone receptor-positive and HER2-negative breast cancer. Across all subtypes, a positive correlation was observed between tumor stiffness and size, while the TIL level showed a significant negative correlation. The TIL level was the only parameter to correlate with low tumor stiffness consistently and significantly, which was further confirmed by linear regression.

**Abstract:**

***Background:*** We aimed to elucidate the clinical significance of tumor stiffness across breast cancer subtypes and establish its correlation with the tumor-infiltrating lymphocyte (TIL) levels using shear-wave elastography (SWE). ***Methods:*** SWE was used to measure tumor stiffness in breast cancer patients from January 2016 to August 2020. The association of tumor stiffness and clinicopathologic parameters, including the TIL levels, was analyzed in three breast cancer subtypes. ***Results***: A total of 803 patients were evaluated. Maximal elasticity (E_max_) showed a consistent positive association with an invasive size and the pT stage in all cases, while it negatively correlated with the TIL level. A subgroup-specific analysis revealed that the already known parameters for high stiffness (lymphovascular invasion, lymph node metastasis, Ki67 levels) were significant only in hormone receptor-positive and HER2-negative breast cancer (HR + HER2-BC). In the multivariate logistic regression, an invasive size and low TIL levels were significantly associated with E_max_ in HR + HER2-BC and HER2 + BC. In triple-negative breast cancer, only TIL levels were significantly associated with low E_max_. Linear regression confirmed a consistent negative correlation between TIL and E_max_ in all subtypes. ***Conclusions***: Breast cancer stiffness presents varying clinical implications dependent on the tumor subtype. Elevated stiffness indicates a more aggressive tumor biology in HR + HER2-BC, but is less significant in other subtypes. High TIL levels consistently correlate with lower tumor stiffness across all subtypes.

## 1. Introduction

Shear-wave elastography (SWE) in an advanced ultrasound technique used to visualize and quantitatively assess tissue stiffness in kPa by employing the generation of radiation force through pulses from a transducer [1]. SWE shows good performance in differentiation between benign and malignant breast lesions, achieving sensitivities and specificities ranging from 80 to 90% [1,2]. Of its quantitative parameters, the maximal elasticity (E_max_) and elasticity ratio (E_ratio_) stand out as the most reliable for breast lesion diagnosis [3,4]. Within the spectrum of malignant breast lesions, SWE serves as a robust predictive tool. It distinctly differentiates invasive carcinoma from intraductal lesions, with the former exhibiting higher stiffness [5,6].

Beyond the differentiation of benign and malignant lesions, the stiffness of lesions determined by elasticity values derived from SWE has been correlated with various clinicopathologic attributes in invasive breast cancer. Higher stiffness has shown significant correlation with poor prognostic factors, including a larger tumor size, lymphovascular invasion (LVI), a higher histologic grade (HG), lymph node (LN) metastasis, a triple-negative status, and a high Ki67 labeling index (LI) [7,8,9,10,11,12]. From a prognostic perspective, higher stiffness has been linked to worse disease-free survival in patients with early breast cancer [13].

While the existing literature has highlighted the correlation between tumor stiffness and the aggressive biological behavior of breast cancer, it is essential to consider the molecular heterogeneity of the disease. Breast cancer displays varying biological characteristics and trajectories across its subtypes: HR + HER2-BC (hormone receptor-positive, human epidermal growth factor receptor 2-negative breast cancer), HER2 + BC, and triple-negative breast cancer (TNBC). In the current therapeutic landscape, the tumor-infiltrating lymphocyte (TIL) level is a key predictor of treatment outcomes, especially for HER2 + BC and TNBC [14,15]. However, the relationship between TIL levels and tumor stiffness remains unexplored. Furthermore, many past studies have focused on individual immunohistochemical profiles rather than evaluating stiffness across the specific breast cancer subtypes. Given the distinct clinicopathological attributes of each subtype, the clinical impact of stiffness could vary. In this study, we examined the ties between clinicopathological factors—including TIL levels—and tumor stiffness across breast cancer subtypes.

## 2. Materials and Methods

### 2.1. Patient Selection and Clinicopathologic Evaluation

Patients with invasive breast cancer (stages I–III; age  ≥  20 years at the time of surgery) who were treated between January 2016 and August 2020 at Gangnam Severance Hospital according to standard protocols were included in this study. A total of 803 female patients underwent SWE examination before curative resection. The tumor stiffness, measured by SWE and documented as elasticity values, was obtained from the patients’ breast ultrasound exam records prior to their curative surgeries. The clinicopathologic parameters evaluated in each case from the electronic medical records included the age of the patient at initial diagnosis, the tumor size, nuclear grade (NG) and HG based on the Nottingham grading system [16], level of TIL, pathologic tumor stage, LVI, LN metastasis, percentage and the extent of intraductal carcinoma component, and the status of estrogen receptor (ER), progesterone receptor (PR), HER2, and Ki67 LI.

### 2.2. Elastography

Breast US examinations were conducted by one of four radiologists with 5–10 years of experience, utilizing the Aixplorer US system (SuperSonic Imagine, Aix-en-Provence, France) equipped with a 4–15 MHz linear array transducer. Investigators were provided with clinical and mammographic results during the breast US exam. Following grayscale ultrasound imaging, Shear Wave Elastography (SWE) images were captured statically for breast masses slated for biopsy or surgical resection. The system’s integrated region of interest (ROI) (Q-box, SuperSonic Imagine) was configured to encompass the lesion and surrounding normal tissue, presenting a grayscale image overlaid with a translucent color map. The tissue stiffness was represented from dark blue (indicating the lowest stiffness) to red (indicating the highest stiffness) within a range of 0–180 kPa (Appendix A). Black regions in the SWE image indicated areas where no shear waves were detected. An investigator placed a fixed 2 × 2 mm ROI in the most rigid part of the lesion, encompassing the immediately adjacent tissue or halo. Elasticity measurements were taken using the average, maximum, and minimum elasticity values, as well as the elasticity ratio, which compares values with adjacent fat tissue.

### 2.3. Pathologic Review of Cases

The histology slides were reviewed by two breast pathologists (YL and YJC). The tumor–stroma ratio (TSR) is defined as the density of tumor cells within the tumor area [17]. A nuclear positivity of 1% or higher was considered positive for ER and PR [18]. The interpretation of HER2 immunohistochemistry was based on the 2018 American Society of Clinical Oncology/College of American Pathologists [19]. Only strong and circumferential membranous HER2 immunoreactivity (3+) was considered positive. Otherwise, 0 or 1+ HER2 staining was considered negative. Cases with equivocal HER2 expression (2+) underwent further evaluation using silver in situ hybridization to evaluate HER2 gene amplification.

### 2.4. Evaluation of TIL

The TIL level was evaluated according to the guidelines suggested by the International TIL Working Group [20]. Briefly, among the total intratumoral stromal area, the percentage of the space occupied by mononuclear inflammatory cells, including lymphocytes and plasma cells, was measured. The tumor area was defined by the boundaries of invasive tumor cells. Polymorphonuclear leukocytes, granulocytes, dendritic cells, and macrophages were excluded from the scoring. Areas beyond the invasive tumor border, such as the periphery of the intraductal component and normal lobules area, were also excluded. Within the tumor border, TILs located in extensive fibrosis, crush artifacts, necrosis, and regressive hyalinization were not considered in the measurement. A comprehensive assessment of the average TIL score was reported as a percentage.

### 2.5. Statistical analysis

Analyses were conducted using R software (https://www.r-project.org (accessed on 18 November 2023); version 4.3.1). In this study, the continuous variables of clinicopathologic parameters were analyzed using Wilcoxon tests, and pairwise Wilcoxon tests with Bonferroni correction were applied for tumor subtype comparisons. As the distribution of the patient number across the tumor subtype was uneven, we chose Wilcoxon tests as an alternative to the *t*-test. For further comparisons across multiple tumor subtypes, we employed pairwise Wilcoxon tests with a Bonferroni correction. This approach was chosen to manage the increased risk of type I errors (false positives) that occurs when conducting multiple comparisons.

The optimal cutoff values distinguishing low and high tumor elasticity were established by analyzing the receiver operating characteristic (ROC) curves, which were plotted based on the relationship between tumor elasticity and LN metastasis. Categorical variables were evaluated with Pearson’s Chi-squared test. Correlations between stiffness and the clinicopathologic parameters were visualized using the ggcorrplot package.

Our study investigated the relationship between tumor elasticity and clinicopathologic parameters using linear and logistic regression analyses. For logistic regression, we categorized tumor elasticity into low and high groups based on a cutoff that was determined using ROC curves, yielding odds ratios (ORs) to assess the impact of elasticity levels on binary clinicopathologic outcomes. In linear regression, we used the continuous values of tumor elasticity to explore the correlations with continuous clinicopathologic parameters, calculating β coefficients to quantify the relationships’ strength and direction. The results from both analyses were summarized in forest plots, displaying point estimates (β coefficients or ORs) along with 95% confidence intervals (CIs). Statistical significance was set at *p* < 0.05.

## 3. Results

### 3.1. Basal Characteristics of Patients

A total of 803 patients with breast cancer were analyzed (Figure 1). HR + HER2-BC was the most predominant subtype (79.5%, n = 628), followed by HER2 + BC (12.8%, n = 103) and TNBC (9.0%, n = 72). The clinicopathologic characteristics of the patients are shown in Table 1. The majority of HR + HER2-BC cases presented with intermediate NG (86.9%) and HG II (69.1%). In contrast, HER2 + BC and TNBC showed a higher prevalence of high NG. Notably, HG III was especially pronounced in TNBC, accounting for 59.7%. LN metastasis was more frequent in HR + HER2-BC (21.2%) than in HER2 + BC (13.6%) and TNBC (9.7%). Ki67 LI was highest in TNBC, followed by HER2 + BC and HR + HER2-BC (*p* < 0.001). The mean TIL level was significantly higher in HER2 + BC and TNBC than HR + HER2-BC (*p* < 0.001). The mean elasticity values demonstrated no significant differences across tumor subtypes.

### 3.2. Clinicopathologic Impact of Tumor Stiffness in Different Subtype of Breast Cancer

We determined three cutoff values for each tumor subtype: the mean, the median, and a value derived from the ROC curves related to LN metastasis for E_mean_, E_min_, and E_max_ (Table 2). Using these cutoff values, the patients were divided into two low- and high-stiffness groups. Given the nine cutoffs used for each elasticity value, distinct significance was observed in various clinicopathologic parameters across each tumor subtype (Figure 2). In the HR + HER2-BC subtype, both the mean and median E_max_ showed significant differences in 9 of the 12 parameters. In the HER2 + BC subtype, comparisons based on the cutoff values of the median E_mean_, the ROC-derived cutoff from E_mean_, the median E_max_, and the ROC-derived cutoff from E_max_ revealed significant differences between the two groups in six parameters. For TNBC, the mean E_max_ cutoff value demonstrated significant differences between the groups for four parameters. Parameters such as the total size, invasive size, LVI, and pT stage exhibited consistent and significant differences between the stiffness groups. In HER2 + BC, the invasive size, HG, and pT stage showed notable differences between the stiffness groups. Using the cutoff values for E_mean_ and E_max_, the TIL level exhibited a significant difference between the stiffness groups. For TNBC, size-related factors, especially the total size, invasive size, and pT stage, showed significant differences between the stiffness groups based on the ROC-derived cutoffs. The TIL level exhibited significant differences between the stiffness groups using the E_mean_ and E_max_ cutoffs, except for when using the ROC-derived cutoffs.

For each tumor subtype, the cutoff of the elasticity values associated with the largest number of significant parameters was averaged to determine the cutoff for distinguishing between the low- and high-stiffness groups. In the HR + HER2-BC subtype, the mean E_max_ and median E_max_ were averaged, establishing 173.0 kPa as the cutoff. For the HER2 + BC subtype, the average of median E_mean_, the ROC-derived threshold from E_mean_, median E_max_, and the ROC-derived threshold from E_max_ was taken, setting the cutoff at 133.0 kPa. In the case of TNBC, the mean E_max_ value of 172.0 kPa was selected as the cutoff.

### 3.3. Subgroup Analysis between Low- and High-Stiffness Groups

Using the predetermined cutoff values, we compared the low- and high-stiffness groups within each tumor subtype based on E_max_ values (Table 3). In all tumor subtypes, three parameters—the invasive size, pT stage, and TIL level—consistently differed between stiffness groups. Specifically, tumors in the high-stiffness group exhibited a larger invasive size, higher pT stage, and lower TIL level.

### 3.4. Correlation Analysis of Tumor Stiffness and Clinicopathologic Parameters

Across all tumor subtypes, the invasive size consistently exhibited a significant positive correlation with stiffness, while the TIL level showed a significant negative trend, except for E_min_ in HR + HER2-BC (Figure 3). Specifically, in HR + HER2-BC, both the total and invasive sizes were positively correlated with stiffness. Ki67 LI was positively associated with E_max_ and E_min_, while the proportion of DCIS and TIL demonstrated a negative relationship with E_mean_ and E_max_. For the HER2 + BC subtype, the invasive size and TSR displayed positive correlations with stiffness, in contrast to the negative associations observed with the proportion of DCIS and TIL level. In TNBC, while the TIL level consistently revealed a significant negative relationship with stiffness, the total size was positively correlated with E_mean_ and E_max_. Information about the Pearson correlation coefficient (*r*) and *p*-value is provided in Appendix A.

### 3.5. Predictive Clinicopathologic Parameters for High Stiffness

Following the logistic regression analysis, only the TIL level turned out to be an independent factor associated with reduced odds of E_max_ across all subtypes (Figure 4). When examining all cases as a whole, a low TIL level (OR 0.981, 95% CI 0.973–0.988), the presence of LVI (OR 1.773, 95% CI 0.128–2.789), and a high pT stage (OR 1.828, 95% CI 1.081–3.090) were identified as independent predictors of a high E_max_. For the HR + HER2-BC, the invasive size (OR 1.590, 95% CI 1.063–2.376) and TIL level (OR 0.988, 95% CI 0.978–0.998) demonstrated significant ORs in predicting a high and low E_max_, respectively. As seen in HR + HER2-BC, the invasive size (OR 5.437, 95% CI 1.429–20.690) and TIL level (OR 0.977, 95% CI 0.961–0.993) significantly predicted a high and low Emax in HER2 + BC. In TNBC, only the TIL level served as a significant negative predictor for a high E_max_ (OR 0.961, 95% CI 0.940–0.983). Detailed values from the regression analyses can be found in Appendix A. Furthermore, the linear regression analysis underscored that the TIL level is the consistent independent predictor for a decreased E_max_ across all subtypes (Appendix A).

## 4. Discussion

In this study, we reinforced the established findings and elucidated the implications of stiffness across various breast cancer subtypes. Moreover, to our knowledge, this is the first study highlighting the strong correlation between TIL levels and reduced tumor stiffness in breast cancer. Our findings validated the significant association of the invasive size with elevated stiffness. In contrast to the invasive size, the proportion of DCIS presented a significant OR for decreased stiffness at the univariate level in both HR + HER2-BC and HER2 + BC. Since DCIS is a non-invasive component that does not evoke the desmoplastic reaction, a tumor with a higher proportion of DCIS might be less stiff than similar-sized tumors with lower DCIS proportions. Previous studies have repeatedly linked Ki67 LI with stiffness [9,11,21,22]. In our research, the multiple linear regression analysis identified a significant positive correlation between Ki67 LI and E_max_, but this was exclusive to the HR + HER2-BC subtype (β 0.43, 95% CI 0.026–0.837, Appendix A). Contrastingly, the univariate logistic regression analysis revealed only a subtle trend associating Ki67 LI with increased stiffness. Given that HR + HER2-BC represents the predominant subtype of breast cancer, it is plausible that the outcomes of previous studies may have been skewed by its characteristics. We verified that the parameters for high stiffness used in preceding studies, such as the tumor size, LVI, LN metastasis, and the tendency towards higher Ki67 LI, were predominantly observed in HR + HER2-BC. These findings remained largely consistent when analyzing all tumors together. However, a more detailed subgroup analysis revealed unique attributes inherent to each subtype. Only a large invasive size and low TIL level were correlated with increased stiffness in HR + HER2-BC and HER2 + BC. TNBC displayed the fewest associations; only the TIL level demonstrated significant OR, with its negative correlation also uniquely identified in the multiple linear regression analysis (Appendix A).

Elevated stiffness in cancer lesions mirrors the invasive nature of cancer cells and their interactions with the stroma. When invasive cancer cells penetrate the basement membrane, they trigger a stromal reaction known as the desmoplastic reaction [23]. The reconstitution of extracellular matrix, particularly by fibroblasts, changes stromal fibers to tenascin and fibronectin, rendering the stroma denser and more rigid, thereby facilitating cancer invasion [24]. In contrast to the typically tight cellular adhesions found in epithelial and stromal cells, TILs display loose adhesions, allowing them to effectively navigate towards target lesions [25,26]. Consequently, an elevated TIL level might disrupt varied cellular adhesions, leading to diminished tumor stiffness. This supports our observation of a consistent negative correlation between the TIL level and stiffness.

The TIL level, in solid tumors including breast cancer, is perceived as an indicator of anti-tumor immunity [27], predicting the response to neoadjuvant chemotherapy (NAC) across all subtypes [15]. NAC is now the standard treatment option in most early HER2 + BC and TNBC patients [28]. Particularly in HER2 + BC and TNBC, the TIL level predicts the long-term survival outcome after NAC as well as the treatment response [29,30,31]. A few antecedent studies have explored the link between stiffness and the response to NAC in breast cancer, albeit with a limited cohort [32,33,34,35]. Their primary emphasis has been on the pre- and post-NAC stiffness alterations, bypassing the significance of TIL levels. Recently, our team validated the importance of TIL levels in HER2 + BC in the NAC setting [36]. In addition, we investigated the association between tumor stiffness and the response to NAC in breast cancer, which showed lower elasticity correlated with a better response to NAC as well as higher TIL levels [37].

This study, despite its insights, has limitations, including the unequal subtype distribution and the unexplored connection between clinical outcomes and tumor stiffness. Still, each subtype retained a representative sample size. Our findings suggest that tumor stiffness measured by SWE could serve as a non-invasive biomarker considering the potential ability of lower stiffness to predict elevated TIL levels in breast cancer, which has not been reported so far. Exploring further subtype-specific implications, our analysis reveals that in HR + HER2-BC, where TIL levels are usually low and upfront surgery is common, stiffness could serve as a non-invasive predictor of aggressive tumor behavior. For HER2 + BC and TNBC, the emphasis shifts to TIL levels, where reduced stiffness may indicate higher TIL levels, reflecting their ability to predict responses to NAC and their prognostic significance for survival outcomes.

## 5. Conclusions

In conclusion, our comprehensive analysis across a large cohort segmented by tumor subtype illuminates the distinct clinical implications of tumor stiffness in breast cancer. Notably, most previously recognized poor prognostic factors related to higher tumor stiffness were primarily observed in HR + HER2-BC. Meanwhile, a consistent inverse correlation emerged between TIL levels and tumor stiffness across subtypes, highlighting their potential predictive and prognostic roles, especially in HER2 + BC and TNBC in the context of NAC. These findings pave the way for future research exploring how integrating tumor stiffness and TIL levels into existing diagnostic and treatment frameworks could enhance personalized treatment strategies and improve patient outcomes.

## Figures and Tables

**Figure 1 cancers-16-01254-f001:**
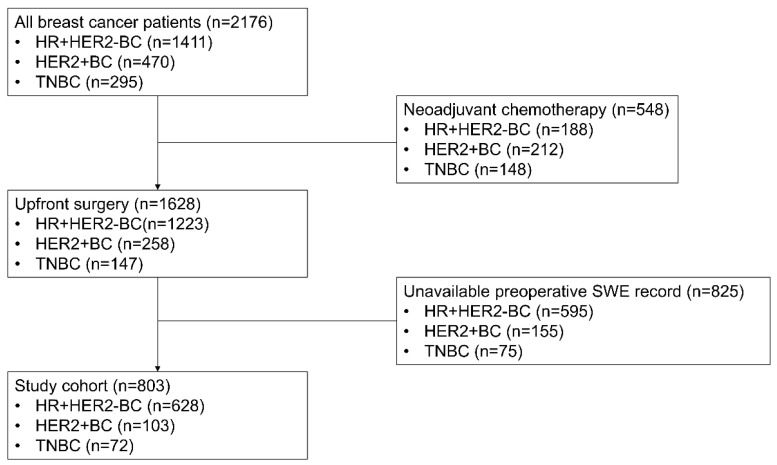
Study flowchart with inclusion and exclusion criteria. HR + HER2-, hormone receptor-positive, HER2-negative; BC, breast cancer; TNBC, triple-negative breast cancer; SWE, shear-wave elastography.

**Figure 2 cancers-16-01254-f002:**
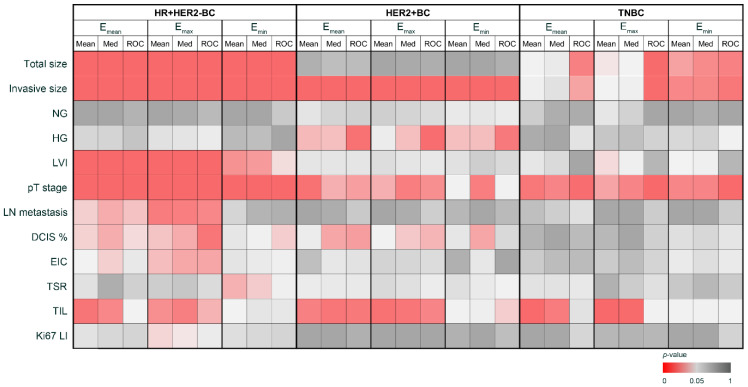
Clinicopathologic implication of tumor stiffness in different tumor subtypes. Med, median; ROC, ROC-derived threshold.

**Figure 3 cancers-16-01254-f003:**
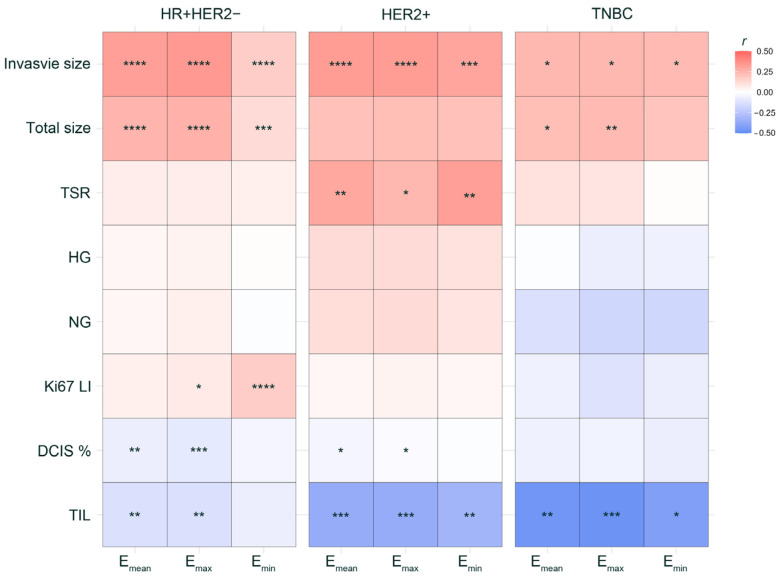
Correlation of tumor stiffness with clinicopathologic parameters in different tumor subtypes. * *p* < 0.05, ** *p* < 0.01, *** *p* < 0.001, **** *p* < 0.0001; *r*, Pearson correlation coefficient.

**Figure 4 cancers-16-01254-f004:**
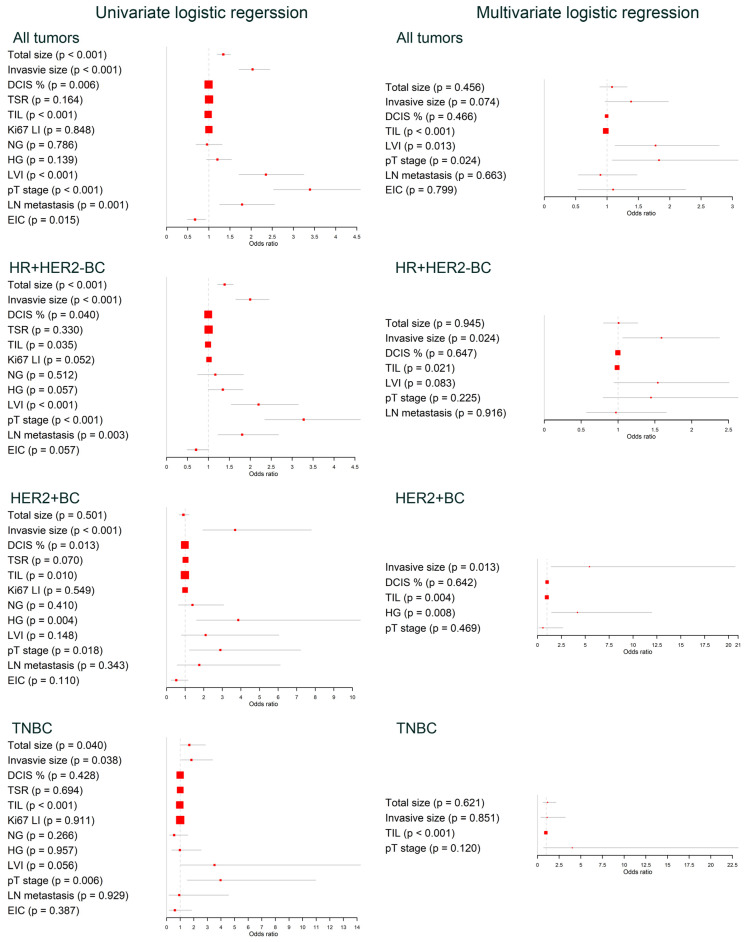
Forest plots of logistic regression analysis assessing the association between tumor stiffness and clinicopathologic parameters across different tumor subtypes.

**Table 1 cancers-16-01254-t001:** Basal clinicopathologic characteristics of patients.

	HR + HER2-BC (n = 628)	HER2 + BC (n = 103)	TNBC (n = 72)	*p*
Age, years (median, IQR)	51 (45–60)	55 (48–60)	54 (46–61)	0.081
Histologic diagnosis, n (%)				0.002
Invasive ductal carcinoma	529 (84.2)	99 (96.1)	61 (84.7)	
Invasive lobular carcinoma	37 (5.9)	1 (1.0)	2 (2.8)	
Mixed invasive ductal carcinoma	15 (2.4)	2 (1.9)	2 (2.8)	
Mucinous carcinoma	27 (4.3)	0 (0.0)	0 (0.0)	
Others	20 (3.2)	1 (1.0)	7 (9.7)	
Total size, cm (mean ± SD)	2.2 ± 1.4	2.6 ± 1.3	2.3 ± 1.1	<0.001 *
Invasive sizes, cm (mean ± SD)	1.8 ± 1.1	1.6 ± 0.8	1.9 ± 0.9	0.110
NG, n (%)				<0.001
Low	8 (1.3)	0 (0.0)	0 (0.0)	
Intermediate	546 (86.9)	44 (42.7)	20 (27.8)	
High	74 (11.8)	59 (57.3)	52 (72.2)	
HG, n (%)				<0.001
I	159 (25.3)	3 (2.9)	0 (0.0)	
II	434 (69.1)	73 (70.9)	29 (40.3)	
III	35 (5.6)	27 (26.2)	43 (59.7)	
LVI, n (%)				0.052
Absent	449 (71.5)	82 (79.6)	59 (81.9)	
Present	179 (28.5)	21 (20.4)	13 (18.1)	
pT stage				<0.001
1	429 (68.3)	70 (68.0)	38 (52.8)	
2	186 (29.6)	33 (32.0)	34 (47.2)	
3	13 (2.1)	0 (0.0)	0 (0.0)	
LN metastasis, n (%)				0.019
Absent	494 (78.8)	89 (86.4)	65 (90.3)	
Present	133 (21.2)	14 (13.6)	7 (9.7)	
DCIS %	19.6 ± 23.5	28.3 ± 308	17.1 ± 25.9	0.003 **
EIC				<0.001
Negative	468 (74.5)	63 (61.2)	55 (76.4)	
Positive	160 (25.5)	40 (38.8)	17 (23.6)	
TSR, % (mean ± SD)	51.5 ± 30.0	61.0 ± 23.4	60.6 ± 26.3	0.321
TIL, % (mean ± SD)	12.8 ± 16.8	37.9 ± 33.2	36.9 ± 31.1	<0.001 ***
Ki67 LI, % (mean ± SD)	12.1 ± 14.2	31.2 ± 18.7	51.1 ± 28.8	<0.001 ****
Stiffness parameters				
E_ratio_ (mean ± SD)	12.7 ± 10.7	13.7 ± 14.2	16.2 ± 20.3	0.366
E_mean_, kPa (mean ± SD)	150.5 ± 65.7	142.0 ± 75.3	151.8 ± 65.8	0.330
E_max_, kPa (mean ± SD)	172.8 ± 72.7	160.0 ± 80.7	171.0 ± 72.2	0.194
E_min_, kPa (mean ± SD)	119.1 ± 85.9	108.7 ± 65.1	122.0 ± 63.4	0.220

Pairwise comparisons adjusted using the post hoc Bonferroni method: * HR + HER2-BC vs. HER2 + BC, *p* < 0.001; HER2 + BC vs. TNBC, *p* = 0.382; HR + HER2-BC vs. TNBC, *p* = 0.455. ** HR + HER2-BC vs. HER2 + BC, *p* = 0.066; HER2 + BC vs. TNBC, *p* = 0.003; HR + HER2-BC vs. TNBC, *p* = 0.059. *** HR + HER2-BC vs. HER2 + BC, *p* < 0.001; HER2 + BC vs. TNBC, *p* = 1.000; HR + HER2-BC vs. TNBC, *p* < 0.001. **** HR + HER2-BC vs. HER2 + BC, *p* < 0.001; HER2 + BC vs. TNBC, *p* < 0.001; HR + HER2-BC vs. TNBC, *p* < 0.001. IQR, interquartile range; SD, standard deviation; EIC, extensive intraductal component (>25%).

**Table 2 cancers-16-01254-t002:** Cutoff of elasticity values and thresholds from ROC curves for lymph node metastasis.

	HR + HER2-BC			HER2 + BC			TNBC		
	Mean	Median	ROC Threshold	Mean	Median	ROC Threshold	Mean	Median	ROC Threshold
E_mean_, kPa	151.9	146.8	134.6	142.7	122.9	121.7	152.1	147.2	140.2
E_min_, kPa	119.0	111.5	121.3	108.6	95.4	110.1	121.2	114.1	100.5
E_max_, kPa	173.8	172.1	178.0	160.5	145.4	141.8	172.0	161.0	150.4

**Table 3 cancers-16-01254-t003:** Comparison between low- and high-stiffness group in each tumor subtype.

	HR + HER2- BC (n = 628) Cutoff E_max_ = 173.0 kPa			HER2 + BC (n = 103) Cutoff E_max_ = 133.0 kPa			TNBC (n = 72) Cutoff E_max_ = 172.0 kPa		
	Low E_max_	High E_max_	*p*	Low E_max_	High E_max_	*p*	Low E_max_	High E_max_	*p*
Total size, cm (mean ± SD)	1.9 ± 1.3	2.5 ± 1.4	<0.001	2.7 ± 1.5	2.5 ± 1.1	0.512	2.1 ± 1.0	2.6 ± 1.1	0.031
Invasive size, cm (mean ± SD)	1.4 ± 0.9	2.2 ± 1.2	<0.001	1.3 ± 0.7	1.9 ± 0.7	<0.001	1.7 ± 1.0	2.2 ± 0.8	0.028
NG, n (%)			0.785			0.532			0.394
1	5 (1.6)	3 (1.0)		0 (0.0)	0 (0.0)		0 (0.0)	0 (0.0)	
2	280 (87.2)	266 (86.6)		23 (46.9)	21 (38.9)		9 (22.5)	11 (34.4)	
3	36 (11.2)	38 (12.4)		26 (53.1)	33 (61.1)		31 (77.5)	21 (65.6)	
HG, n (%)			0.134			0.004			1.000
I	92 (28.7)	67 (21.8)		2 (4.1)	1 (1.9)		0 (0.0)	0 (0.0)	
II	213 (66.4)	221 (72.0)		41 (83.7)	32 (59.3)		16 (40.0)	13 (40.6)	
III	16 (5.0)	19 (6.2)		6 (12.2)	21 (38.9)		24 (60.0)	19 (59.4)	
LVI, n (%)			<0.001			0.223			0.093
Absent	254 (79.1)	195 (63.5)		42 (85.7)	40 (74.1)		36 (90.0)	23 (71.9)	
Present	67 (20.9)	112 (36.5)		7 (14.3)	14 (25.9)		4 (10.0)	9 (28.1)	
pT stage, n (%)			<0.001			0.028			0.010
1	261 (81.3)	168 (54.7)		39 (79.6)	31 (57.4)		27 (67.5)	11 (34.4)	
2	58 (18.1)	128 (41.7)		10 (20.4)	23 (42.6)		13 (32.5)	21 (65.6)	
3	2 (0.6)	11 (3.6)		0 (0.0)	0 (0.0)		0 (0.0)	0 (0.0)	
LN metastasis, n (%)			0.004			0.504			1
Absent	268 (83.5)	226 (73.9)		44 (89.8)	45 (83.3)		36 (90.0)	29 (90.6)	
Present	53 (16.5)	80 (26.1)		5 (10.2)	9 (16.7)		4 (10.0)	3 (9.4)	
DCIS, % (mean ± SD)	21.5 ± 25.6	17.5 ± 21.0	0.034	36.3 ± 34.1	21.0 ± 25.6	0.012	19.3 ± 27.0	14.4 ± 24.6	0.426
EIC, n (%)			0.062			0.160			0.556
Absent	228 (71.0)	239 (77.9)		26 (53.1)	37 (68.5)		29 (72.5)	26 (81.3)	
Present	93 (29.0)	68 (22.1)		23 (46.9)	17 (31.5)		11 (27.5)	6 (18.8)	
TSR, % (mean ± SD)	50.1 ± 29.2	52.8 ± 30.6	0.331	55.3 ± 26.4	64.7 ± 20.6	0.083	59.3 ± 26.7	62.2 ± 26.3	0.700
TIL, % (mean ± SD)	14.3 ± 18.2	11.3 ± 15.0	0.028	46.8 ± 35.4	29.7 ± 29.1	0.009	49.4 ± 29.9	21.4 ± 35.4	<0.001
Ki67 LI, % (mean ± SD)	11.0 ± 13.5	13.2 ± 14.8	0.051	32.4 ± 18.5	30.2 ± 19.0	0.552	51.4 ± 27.0	50.7 ± 24.6	0.911

In the HR + HER2-BC subtype, the high-stiffness group showed a higher frequency of LVI (*p* < 0.001) and LN metastasis (*p* = 0.004). Such trends were not observed in the high-stiffness groups of other subtypes. Additionally, the proportion of ductal carcinoma in situ (DCIS) was significantly lower in the high-stiffness groups of both HR + HER2-BC (*p* = 0.003) and HER2 + BC (*p* = 0.012). For the HER2 + BC subtype, tumors of the high-stiffness group had a more frequent HG III (*p* = 0.004).

## Data Availability

The data presented in this study are available upon request from the corresponding author.

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
