# Peer review of "Tumor-Infiltrating Lymphocyte Level Consistently Correlates with Lower Stiffness Measured by Shear-Wave Elastography: Subtype-Specific Analysis of Its Implication in Breast Cancer"

_cancers, 2024, doi:10.3390/cancers16071254_

Round 1

Reviewer 1 Report

Comments and Suggestions for Authors

The present study by Eun and colleagues demonstrated a potential connection between tumor stiffness, as measured by shear-wave elastography (SWE), and tumor-infiltrating lymphocyte (TIL) levels in breast cancer patients. Patients were categorized into 3 subgroups based on their clinical phenotype. This study analyzed clinical data of 803 patients, exploring how different breast cancer subtypes exhibit varying clinical implications based on tumor stiffness and TIL levels. The authors show that higher tumor stiffness is associated with more aggressive tumor features, particularly in hormone receptor-positive, HER2-negative breast cancers (HR+HER2-BC), while higher TIL levels are consistently linked with lower tumor stiffness across all subtypes. This research could provide new insights into the diagnostic and therapeutic approaches for breast cancer based on mechanical properties of tumors and immune infiltration levels. Overall, the manuscript provides interesting findings that could have potential clinical implications in breast cancer treatments. However, there are several areas where improvements could significantly enhance the clarity, robustness, and impact of the research.

Major Comments:

Statistical Analysis: While the study employs a comprehensive statistical approach, the justification for the selection of certain statistical tests over others is lacking. A more detailed explanation, including the assumptions of each statistical test and how they are met with the current data, would enhance the reader's understanding and trust in the findings.

Clinical Relevance: The manuscript could further elaborate on the clinical implications of the findings. Specifically, how the observed correlations between TIL levels and tumor stiffness could impact current diagnostic or therapeutic approaches in breast cancer management. Additional insights into the distinction between different breast cancer subtypes and the implications of these differences would be beneficial.

The discussion could be enhanced by a more thorough comparison with existing literature, particularly recent studies that have explored similar themes. This would help to situate the study within the current scientific landscape and highlight its novel contributions.

Minor Comments:

All the legends of figures and tables have repetitive abbreviations (Line 152-154, 165, 191-192, 194-196, 214-218, 240, 260-263). In general, the first time you use an abbreviation, write out the term in full, followed by the abbreviation in parentheses. After this, you can use the abbreviation alone. Alternatively, it is advisable to consider including a list of abbreviations at the beginning or end.

Line 112-114 doesn’t seem to be a complete sentence.

With the suggested revisions, I believe the manuscript could provide valuable insights into the relationship between tumor stiffness and TIL levels, thereby influencing future research and clinical practices in breast cancer management.

Please address these comments in your manuscript revision to enhance the clarity and impact of your study.

Comments on the Quality of English Language

Overall, the manuscript is proficiently written in scientific English with only minor editing needed, and the language used is suitable.

Reviewer 2 Report

Comments and Suggestions for Authors

I have completed the review of the submission titled "Tumor-infiltrating lymphocyte level consistently correlates with lower stiffness measured by shear-wave elastography: subtype-specific analysis for its implication in breast cancer" by Eun et.al. The manuscript is well-written and effectively communicates the research findings. The methodology is sound, and the research question is clearly defined. The findings are significant and make a valuable contribution to the field. I am pleased to inform you and the author that I recommend acceptance of the manuscript without any revisions.

Author Response

We would like to express our sincere gratitude to Reviewer 2 for his/her thorough review and positive feedback on our manuscript. We are deeply appreciative of the reviewer's recognition of the significance and contribution of our research findings to the field.